# A Surface Potential Model for Metal-Oxide-Semiconductor Transistors Operating near the Threshold Voltage

**Hwang-Cherng Chow \*, Bo-Wen Lee, Shang-Ying Cheng, Yung-Hsuan Huang and Ruey-Dar Chang \***

Department of Electronics Engineering, Chang Gung University, 259 Wenhua 1st Road, Kweishan, Taoyuan 33302, Taiwan; d0827101@cgu.edu.tw (B.-W.L.); m0727106@cgu.edu.tw (S.-Y.C.); m0427103@cgu.edu.tw (Y.-H.H.)

\* Correspondence: hcchow@mail.cgu.edu.tw (H.-C.C.); crd@mail.cgu.edu.tw (R.-D.C.)

**Abstract:** Device physics and accurate transistor modeling are necessary to reduce the operating voltage near the threshold for power-constrained circuits. Conventional device modeling for metal-oxide-semiconductor (MOS) transistors focuses on operations in either strong or weak inversion regimes, and the electrostatics at gate biases near the threshold voltage is rarely studied. This research proposed an analytical model to describe the distribution of the surface potential along the channel for near-threshold operation. Numerical device simulations were also performed to investigate the electrostatics near the threshold voltage. The numerical simulation with constant carrier mobility showed an overshoot in the transconductance due to decay of the lateral electric field with gate bias. The decay of the lateral electric field was predicted by the proposed analytical surface potential model which considered widening the channel length with flooding of the inversion carriers in the channel and gate overlap regions. The channel length widening effect saturated as the gate bias further increased. Therefore, evident transconductance overshoot was observed near the threshold voltage in short-channel devices.

**Keywords:** surface potential; near-threshold operation; field-effect transistor; transconductance

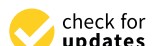



## 1. Introduction

Power consumption has become a major constraint in the development of sensors in the internet of things. Near-threshold computing was proposed to optimize the energy efficiency of circuits [1] contrary to minimizing the operation time delay. The trade-off between performance and robustness is crucial for low-power circuits. The operating voltage of field-effect transistors is carefully adjusted based on the impact of process variations on the circuit performance [2]. A deep understanding of device physics and accurate transistor modeling is necessary to further reduce the operating voltage near the threshold voltage. Conventional device modeling for metal-oxide-semiconductor field effect transistors (MOSFETs) focuses on operations in either strong or weak inversion regimes. In a weak inversion, the transistor channel potential is dominated by space charges. Strong inversion results in a charge sheet between the source and drain regions at gate voltages that are much higher than that of the threshold voltage. Unfortunately, the electrostatics at gate biases near the threshold voltage has rarely been investigated. A compact model was developed using a continuous function to describe the transition behavior between different operating regimes [3]. However, the evolution of electrostatics has not been considered.

Transconductance and channel length are fundamental transistor parameters related to carrier velocity [4,5], magnetoresistance [6,7] and small-signal analysis. The variations in transconductance and channel length are critical for analog circuit design. The decay of transconductance with increasing gate voltage is usually observed because of carrier mobility degradation caused by surface roughness scattering under a high transverse

electric field [8]. In this study, we performed a numerical device simulation to investigate the transistor behavior near the threshold voltage. An evident transconductance overshoot in short-channel devices was observed when a constant carrier mobility was assigned in the simulation. Conventional device models cannot explain this anomalous phenomenon. A quasi-two-dimensional Poisson's equation has been solved for modeling the drain potential in the saturation region [9] or extracting the threshold voltage in the depletion region [10,11]. We modified the equation to consider the flooding of inversion carriers in the channel for modeling the surface potential near the threshold. A channel widening mechanism was proposed to explain the transconductance overshoot which is not related to the mobility degradation.

## 2. Numerical Device Simulation

A two-dimensional device simulator, MEDICI [12], was used to extract the device characteristics of n-type planar MOSFETs. As shown in Figure 1, the device structure has a gate oxide layer with a thickness of 7 nm on a p-type substrate with a uniform doping concentration of $N_a = 2 \times 10^{17}$ cm$^{-3}$. The source and drain regions were doped with donors at a concentration of $N_d = 10^{20}$ cm$^{-3}$ and the junction depth was approximately 0.2 μm. Portions of the source and drain regions were placed under the gate electrode. A constant electron mobility of 1200 cm$^2$/V-s was assigned to eliminate the mobility degradation factor in the numerical simulation. A simulation was performed for each gate length ($L_g$) to obtain the transconductance according to the $I_d$-$V_g$ curves at $V_d = 0.05$ V.

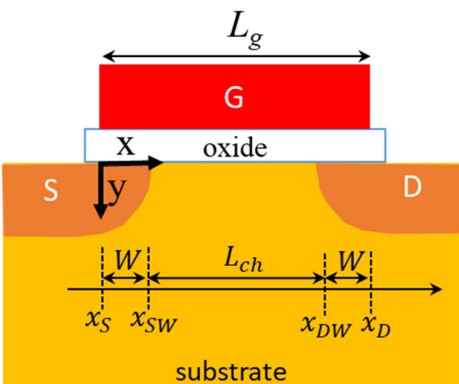

**Figure 1.** Schematic of the MOSFET structure and the definition of regions in this study.

Figure 2 shows the $I_d$-$V_g$ curves for devices with different lengths. The current of the device with $L_g = 0.7$ μm is higher than that with $L_g = 10$ μm. However, the slope of the subthreshold current is similar between the curves for devices with different gate lengths. The threshold voltages of the two devices do not exhibit significant difference. This implies that conventional short-channel effects are not evident in the simulation study for devices with $L_g = 0.7$ μm. The transconductance of field-effect transistors depends on the channel length. Therefore, the simulated transconductance was normalized to $1/L_g$, as shown in Figure 3. The transconductance increases with the formation of inversion carriers in the channel. For the transistor with $L_g = 10$ μm, the transconductance approached a constant value as the gate voltage increased further. The normalized transconductance increased slightly with decreasing gate length. This was caused by a reduction in the channel length owing to encroachment from the source and drain regions. However, the device with $L_g = 0.7$ μm showed an evident transconductance overshoot at the gate bias just above the threshold voltage. The transconductance decreased by more than 10% as $V_g$ increased from 1 to 5 V. Because the carrier mobility in the simulation was constant, the influence of mobility degradation on the transconductance could be neglected.

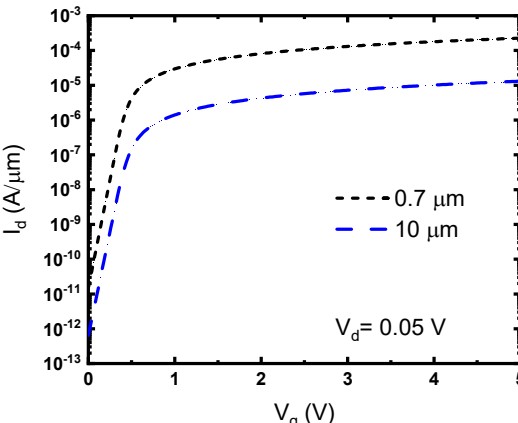

**Figure 2.** Numerical simulation result showing the $I_d$-$V_g$ curves for devices with different gate lengths.

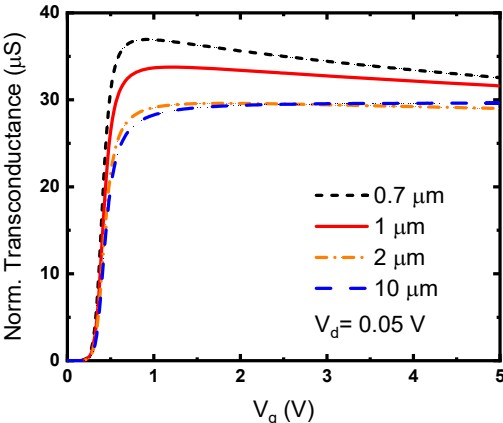

**Figure 3.** Numerical simulation result showing the normalized transconductance as a function of gate bias and length.

The correlation between the physical quantities was examined to determine the mechanism causing the transconductance decay. For a physical quantity M, the incremental change $\Delta M$ responding to $\Delta V_g$ = 0.5 V was extracted from the simulation results with different gate voltages. Current density J and electron concentration n in the middle of the channel were verified for a transistor with $L_g$ = 0.7 μm. Figure 4 shows the depth profiles of the electron concentration as a function of $V_g$. Although both n and J increased with the gate bias, $\Delta n/n$ remains similar while $\Delta J/J$ decayed more than 10% when $V_g$ increased from 1 to 5 V. Clearly, the decay in current density was related to the transconductance overshoot while the amount of inversion carriers in the channel was irrelevant. Figure 5 shows the magnitude of the lateral electric field along the y direction in the middle of the channel at $L_g$ = 0.7 μm. Interestingly, the magnitude of the electric field at $V_g$ = 1 V was larger than that at 5 V, with a difference of approximately 10%. This suggests that the overshoot in the transconductance was caused by a change in the lateral electric field. However, the magnitude of the lateral electric field was much smaller than that of the vertical field. This results in difficulty to further analyze the two-dimensional electric field. Therefore, a surface potential model should be developed.

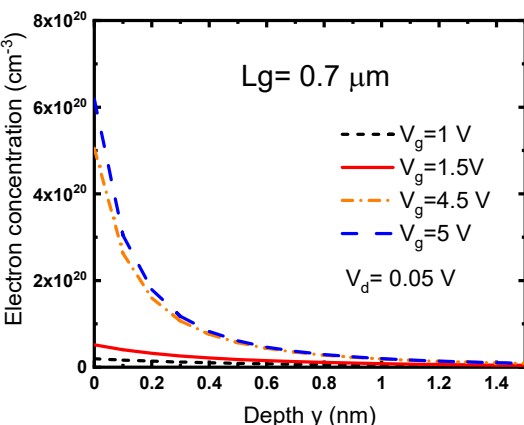

**Figure 4.** Depth profiles of electron concentration in the middle of the channel with a gate length of 0.7 μm obtained from numerical simulation.

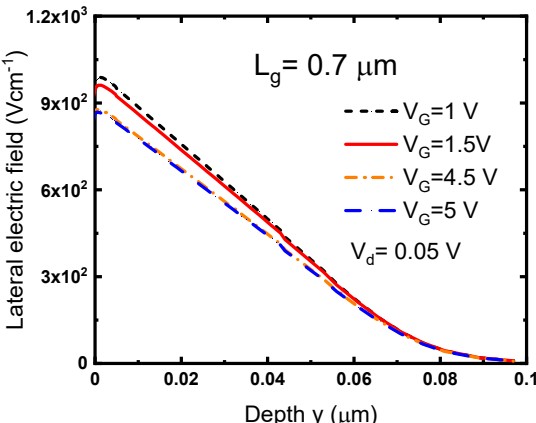

**Figure 5.** Depth distribution of lateral electric field in the middle of the channel with a gate length of 0.7 μm extracted from the result of numerical device simulation.

## 3. Analytical Modeling of Surface Potential

For modeling of the surface potential along the channel at gate biases just above the threshold voltage, the electrostatics in the inversion regime was analyzed based on the gradual channel and charge sheet approximations. As shown in Figure 1, the width of the gate-to-source and gate-to-drain overlap regions is defined as $W = x_{SW} - x_S = x_D - x_{DW}$ for a device with a channel length of $L_{ch} = x_{DW} - x_{SW}$. In a strong inversion, the inversion charge density $Q_i$ along the channel region can be approximated as

$$Q_i = -C_{ox}\left[V_g - V_{fb} - \phi_{si} - V_{ch}(x)\right] - Q_d, \tag{1}$$

where $C_{ox}$ is the capacitance of the gate oxide, $V_{fb}$ is the flat band voltage, $\phi_{si}$ is the inversion surface potential, and $V_{ch}(x)$ is the channel potential [8]. The charge density in the substate depletion layer $Q_d$ is expressed as

$$Q_d = -qN_aX_{dep}, \tag{2}$$

where $X_{dep}$ is the depletion width, which can be assumed to be constant at low $V_d$. Based on the quasi-two-dimensional Poisson's equation [13,14], $Q_i$ can be derived from

$$-Q_i = \frac{\epsilon_sX_{dep}}{n_{sp}}\frac{d^2\phi_s(x)}{dx^2} + C_{ox}\left[V_g - V_{fb} - \phi_s(x)\right] - qN_aX_{dep}, \tag{3}$$

where $\epsilon_s$ is the permittivity of the substate and $n_{sp} = 2$ is adopted for the magnitude of the lateral electric field linearly deceasing into the bulk [15]. Combining Equations (1)–(3) yields

$$\frac{\epsilon_s X_{dep}}{n_{sp}} \frac{d^2 \phi_s(x)}{dx^2} = C_{ox}[\phi_s(x) - \phi_{si} - V_{ch}(x)] \tag{4}$$

with boundary conditions of $\phi_{SW} = \phi_s(x_{SW})$ and $\phi_{DW} = \phi_s(x_{DW})$, where $x = x_{SW}$ and $x = x_{DW}$ are the edges of the source and drain regions, respectively. $V_{ch}(x)$ can be calculated from the channel length $L_{ch}$ as follows:

$$V_{ch}(x) = \frac{\phi_{DW} - \phi_{SW}}{L_{ch}} x' = E_0 x', \tag{5}$$

where $x' = x - x_{SW}$. Equation (4) can be rearranged as

$$\frac{d^2 \phi_s(x')}{dx^2} - K^2 \phi_s(x') = -K^2(E_0 x' + \phi_{si}), \tag{6}$$

where the constant K is defined as

$$K = \sqrt{\frac{C_{ox} n_{sp}}{\epsilon_s X_{dep}}}. \tag{7}$$

The solution of Equation (6) can be obtained as

$$\phi_s(x) = (\phi_{DW} - E_0 x_{DW} - \phi_{si}) \frac{\sin h[K(x - x_{SW})]}{\sin h(KL_{ch})} + (\phi_{SW} - E_0 x_{SW} - \phi_{si}) \frac{\sin h[K(x_{DW} - x)]}{\sin h(KL_{ch})} + E_0 x + \phi_{si}. \tag{8}$$

The conventional definition of the inversion surface potential $\phi_{si}$ was twice the Fermi potential $\phi_f$ regarding the p-type substrate. However, $\phi_{si}$ was considered as a weak function of $V_g$ in this study. Based on one-dimensional analysis of a MOS structure on a p-type substrate, the correlation between $V_g$ and surface potential $\phi_{s0}$ can be expressed as

$$V_g - V_{fb} - \phi_{s0} = \gamma \sqrt{\phi_{s0} + \left(\frac{kT}{q}\right) e^{\frac{q(\phi_{s0} - 2\phi_f)}{kT}}}, \tag{9}$$

where the body factor $\gamma$ is

$$\gamma = \frac{\sqrt{2q\epsilon_s N_a}}{C_{ox}}. \tag{10}$$

According to Equation (9), the surface potential in the strong inversion regime can be approximated as

$$\phi_{si} = 2\phi_F + \left(\frac{kT}{q}\right) \ln\left\{\frac{q}{kT}\left[\left(\frac{V_g - V_{fb} - \phi_*}{\gamma}\right)^2 - \phi_*\right]\right\} \tag{11}$$

similar to a previous study [16].

To determine $\phi_*$, $\phi_{s0}$ in Equation (9) is replaced by $\phi_*$ with the assumption of

$$\phi_* = 2\phi_f + \phi_\alpha + \Delta\phi_*. \tag{12}$$

After simplification of Equation (9), $\Delta\phi_*$ can be obtained as

$$\Delta\phi_* = \frac{\left(\frac{kT}{q}\right) \ln\left[\frac{q}{kT}\left(\frac{V_\delta^2}{\gamma^2} - 2\phi_f - \phi_\alpha\right)\right]}{\left[\left(\frac{\frac{2V_\delta}{\gamma^2} + 1}{\frac{V_\delta^2}{\gamma^2} - 2\phi_f - \phi_\alpha}\right) + 1\right]}, \tag{13}$$

where parameter $V_\delta$ is

$$V_\delta = V_g - V_{fb} - 2\phi_f - \phi_\alpha. \tag{14}$$

Parameter $\phi_\alpha$ was chosen to be $3kT/q$ to optimize $\phi_*$ for an accurate solution of $\phi_{si}$. Figure 6 compares the inversion surface potential regarding the p-type substrate according to Equations (9) and (11). An accurate description of $\phi_{si}$ can be achieved by Equation (11) for $V_g$ values between 0.6 and 5 V.

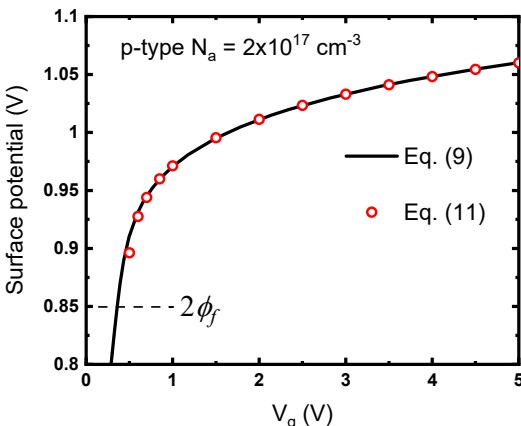

**Figure 6.** Surface potential for a p-type substrate in the strong inversion regime based on Equations (9) and (11).

The surface potentials in the source and drain regions under the gate should also be considered for short-channel characteristics. In an n-type region, the modulation of surface potential $\phi'_{s0}$ by the gate bias can be written as

$$V_g - V'_{fb} - \phi'_{s0} = \gamma' \sqrt{\frac{kT}{q}(e^{\frac{q\phi'_{s0}}{kT}} - 1) - \phi'_{s0}}, \tag{15}$$

where

$$\gamma' = \frac{\sqrt{2q\epsilon_s N_d}}{C_{ox}} \tag{16}$$

and

$$V'_{fb} = V_{fb} - \phi_{bi} = V_{fb} - \frac{kT}{q}\ln\left(\frac{N_d N_a}{n_i^2}\right). \tag{17}$$

For the surface potential $\phi'_{ac}$ in the accumulation regime, Equation (15) can be simplified as follows:

$$\phi'_{ac} = \left(\frac{kT}{q}\right)\ln\left\{\frac{q}{kT}\left[\left(\frac{V_g - V'_{fb} - \phi'_*}{\gamma'}\right)^2 - \phi'_*\right] + 1\right\} \tag{18}$$

where

$$\phi'_* = \frac{V_g - V'_{fb}}{\left(1 + \gamma'\sqrt{\frac{q}{2kT}}\right)}. \tag{19}$$

Figure 7 shows good agreement between the surface potentials calculated using Equations (18) and (15). This indicates that the surface potential in the n-type region can be accurately described using Equation (18) for $V_g$ values ranging from 0 to 5 V. The surface potential was assumed to be governed by the gate bias at the two edges of the gate electrode. For $x = x_S$ at the gate edge near the source side, the surface potential is obtained as follows

$$\phi_s(x_S) = \phi_{Sov} = \phi'_{ac} + \phi_{bi}. \tag{20}$$

Similarly, the surface potential for $x = x_D$ at the gate edge near the drain side can be determined as

$$\phi_s(x_D) = \phi_{Dov} = \phi'_{ac} + \phi_{bi} + V_d. \tag{21}$$

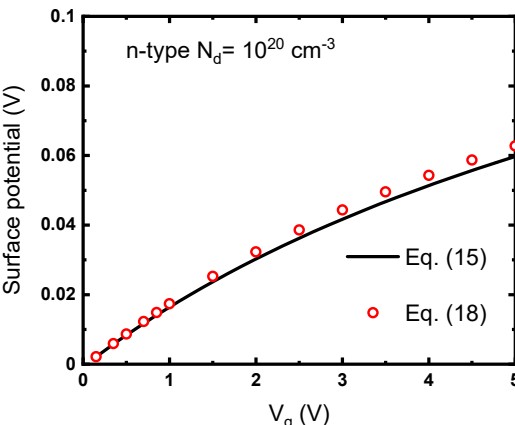

**Figure 7.** Surface potential as a function of the gate bias calculated by Equations (15) and (18) regarding an n-type substrate in the accumulation regime.

The electrostatics in the source and drain overlap regions was dominated by the lateral electric field caused by space charges. When more carriers flooded into the overlapping regions, the lateral electric field decreased. Therefore, the widths of the overlapping region were reduced. Because the drain bias in this study was low, the width and potential drop in the drain overlap region were assumed to be the same as those in the source region. The widths of the overlapping regions were weakly modulated by the gate bias. Thus, $x_{SW}$ and $x_{DW}$ varied with $V_g$. The modulation of the width of the overlapping region can be expressed as

$$W = W_0 \left\{ 1 - \frac{C_{ox}(V_g - V_{fb})}{[Q_{j0} + C_{ox}(V_g - V_{fb})]} \right\}, \tag{22}$$

where $W_0$ represents a parameter associated with the original width and $Q_{j0}$ is the existing carrier density in the overlap regions for the gate bias around the flat band voltage. The depletion behavior in the overlapping regions was complex because of the two-dimensional distributions of source and drain doping. Therefore, empirical hyperbolic functions were adopted to approximate the potential distribution in overlapping regions. For $x_S \leq x \leq x_{SW}$, the distribution of the surface potential can be approximated as

$$\phi_s(x) = \phi_{Sov} - \frac{\Delta\phi_W}{2}\tan h[\kappa\frac{(x - x_S - W/2)}{W/2}] - \frac{\Delta\phi_W}{2}\tan h(\kappa), \tag{23}$$

where $\kappa = 1.1$ is a fitting parameter and $\Delta\phi_W$ can be obtained using

$$\Delta\phi_W = \frac{qN_{eff}}{2\epsilon_s}W^2, \tag{24}$$

with an effective junction doping $N_{eff} = 1.5 \times 10^{16}$ cm$^{-3}$ to describe the evolution of the surface potential in the junction regions. The effective junction doping is lower than that in the numerical simulation because of the strong perturbation of the gate bias on the two-dimensional electrostatics near the junctions.

Similarly, the surface potential in the drain overlap region can be described as

$$\phi_s(x) = \phi_{Dov} + \frac{\Delta\phi_W}{2}\tan h[\kappa\frac{(x - x_D + W/2)}{W/2}] - \frac{\Delta\phi_W}{2}\tan h(\kappa) \tag{25}$$

for $x_{DW} \leq x \leq x_D$.

The surface potential obtained from the analytical model and the MEDICI simulation are shown in Figure 8. The lateral electric field extracted from the surface potential in the middle of the channel is shown in Figure 9. Good agreement was demonstrated between the modeling results and those obtained from the numerical simulation using MEDICI. The gate bias modulates not only the surface potential in the channel but also the potential in the source and drain overlap regions. When more inversion carriers were induced at high gate biases, the influence of the depletion charges on the surface potential decreased. Therefore, the potential decrease in the source and drain regions decreases, whereas the distance between the source and drain edges increases slightly. This causes a widening of the channel length and thus a decrease in the lateral electric field. The widening effect saturated as the gate bias increased further. As a result, the decay in the lateral electric field and the transconductance overshoot were evident near the threshold voltage. The model successfully revealed the mechanism causing the decay of the lateral electric field with the gate bias. However, the two-dimensional effects in the source and drain regions are complex, causing an error between the modeling results and numerical simulation. Widening of the channel length can be ignored in long-channel devices. This explains why the overshoot in the transconductance becomes evident with a decrease in the gate length.

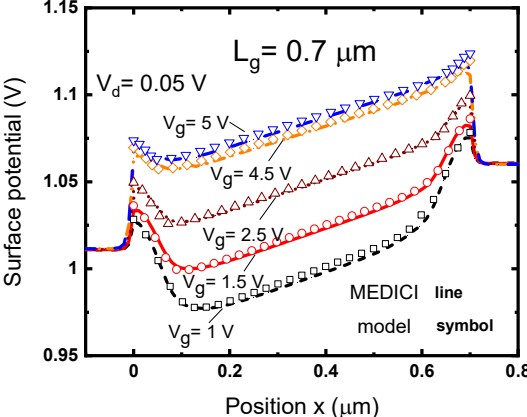

**Figure 8.** Comparison of the distributions of the surface potential along the channel obtained from the analytical model and numerical simulation.

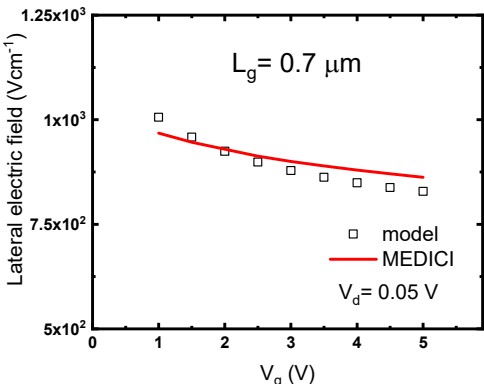

**Figure 9.** Comparison between the magnitude of the lateral electric field in the middle of the channel extracted from the results of numerical simulation and analytical model.

## 4. Discussion

In the numerical simulation, a default model based on Boltzmann statistics was used for calculation of the carriers in the source and drain regions. Other mechanisms related to high-doping effects were not included. This could possibly cause some error in the calculation of the carrier density in real devices. There are different doping layers in

practical MOSFETs. However, nonuniform doping profiles cause more difficulty in solving the electrostatics equations. Simplified numerical simulations, in contrast, can be used to verify analytical models. Thus, simplified models were valid in this study to purely investigate the electrostatics near the threshold voltage. The source and drain regions in the simulation were doped at a concentration of $N_d = 10^{20}$ cm$^{-3}$. Electrons accumulated in these n-type regions. The influence of hole concentration on the surface potential can be ignored. Only electrons and ionized donors were considered in Equation (15). Figure 8 shows the increase of the surface potential at the edges of the gate electrode. The surface potential predicted by the analytical model matches well with that obtained from the numerical simulation. This confirms that Equations (15) and (18) capture the potential change in the n-type source and drain regions.

The change of the transconductance with the gate bias is usually interpreted as a result of carrier mobility degradation. The transconductance overshoot caused by the channel widening can be taken into account by an effective mobility in device models. However, this may introduce some error in circuit simulation because the effective mobility in device modeling is usually independent of the gate length. The transconductance overshoot caused by the channel widening is a function of the gate bias and length. The influence of carrier mobility and channel widening on the transconductance should be separated. The carrier mobility is a strong function of the electric field. However, the electric field and the surface potential mainly depend on the charge distribution which is generally independent of the carrier mobility. The channel widening effect is expected to be similar with different mobility models.

A study was conducted by Dutta et al. [17] to describe the impact of source and drain regions on short-channel effects. They performed numerical simulation to study pure electrostatic effects. This is similar to our approach in this paper. They changed the boundary conditions with and without source and drain regions in the simulation. Their research focused on short-channel effects. No inversion carriers were considered in their study. The built-in potential near the source and drain junctions was affected by the lateral electric field originating from space charges. Thus, the short-channel effect was investigated in the depletion condition for the gate bias below the threshold voltage. Widening of the depletion regions further enhanced the short-channel effect. However, the overshoot in the transconductance was observed at gate biases just above the threshold voltage. The influence of the inversion carriers on the surface potential was taken into account in our study. This is the major difference between our research and that by Dutta et al. [17] for investigating the impact of the source and drain regions on the channel potential.

According to the device physics, when the gate bias is over the threshold voltage, carriers flow into the channel from the source and drain regions. The depletion regions around the source and drain junctions no longer exist near the surface. However, the lateral electric field originating from the space charges still exists. The lateral electric field creates a potential drop near the source and drain junctions, as shown in Figure 8. With more inversion carriers flooding in the channel, the lateral electric field is reduced due to the screening effect of the carriers. The regions affected by the lateral electric field also shrink. Therefore, the channel widening effect occurs. Figure 10 depicts the channel widening effect. The space charges that contribute to the lateral electric field are marked by circles. A similar widening effect in bipolar junction transistors is the Kirk effect [18]. However, the Kirk effect is caused by high-level injection of carriers. The carriers in this case are induced by the gate bias. Equation (22) describes the modulation of the overlapping regions by the gate bias. When the inversion carriers begin to flow into the channel, the widening effect is evident. The widening effect saturates when more inversion carriers are in the channel. Therefore, the shrinkage of the overlapping regions is not significant as the gate bias increases further. Notably, the channel widening effect is not related to the conventional short-channel effect. Figure 2 demonstrates that the short-channel effect is not evident in the device with $L_g = 0.7$ µm. However, the transconductance overshoot in the short-channel device is significant. This is because of the high ratio of the widening

effect in the channel length. In fact, the potential change around the source and drain junctions is similar between devices with $L_g$ = 0.7 and 10 μm. The flooding of inversion carriers is expected to occur in advanced non-planar devices. However, its influence on the widening of the channel length needs further investigation. The formation of inversion carriers in non-planar devices is hard to predict because of strong three-dimensional effects. The channel widening effect only becomes evident when the carrier flooding occurs at the major current conducting path.

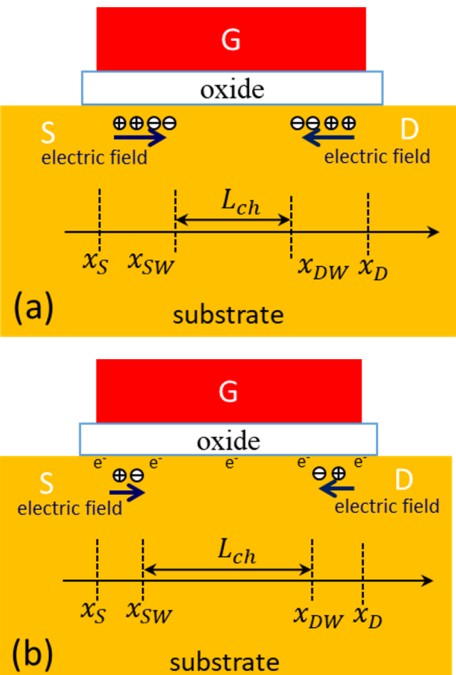

**Figure 10.** Schematic showing the electric field and channel length (**a**) before and (**b**) after the flooding of inversion carriers in the channel.

The channel widening effect occurs at the gate biases just above the threshold voltage. The bias condition is different from that usually used for measuring the channel length either by resistance or capacitance [8]. Strong inversion is necessary to obtain the intrinsic capacitance for channel length measurement. Different gate biases are applied to extract the channel resistance related to the channel length. The change of the channel length with the gate bias is usually treated as a measurement noise. The series resistance contributed from the source and drain regions causes more uncertainty. The dependence of the series resistance on the gate bias increases with the use of lightly doped drain in device structures. It is difficult to observe the variation in the channel length near the threshold voltage. Furthermore, the channel length is usually defined by the metallurgical junctions which depend on the doping profiles. The conventional definition of the channel length is primarily used to control the short channel effect with process variations. In the research, the channel length is related to the potential drop contributed by the space charges near the source and drain junctions. This definition is used to identify the decay in the lateral electric field causing the transconductance overshoot. The electrostatics near the threshold voltage is different from that in other operating regimes.

Conventional approaches to calculate the potential across p-n junctions are based on depletion approximation. In MOSFETs, the depletion regions near the source and drain junctions are modulated by the gate bias. However, in this case, inversion carriers flood into the junctions at gate biases near the threshold voltage. The depletion approximation overestimates the drop of the surface potential around the source and drain junctions. The depletion region deeply extends into the channel region if two-dimensional effects are considered. These effects are not observed in the result obtained from the numerical simulation.

Therefore, the surface potential obtained from the two-dimensional numerical simulation was empirically fitted by hyperbolic functions, as shown in Equations (23) and (25). The modulation of the gate bias is described by Equations (22) and (24). Because of the flooding of inversion carriers, the effective junction doping concentration $N_{eff}$ is much lower than that of the substrate and the source/drain regions in the numerical simulation. This approach catches the change of the surface potential in the overlapping region. However, only the continuity of the surface potential is considered at the edges of the source and drain regions. The continuity of the electric field is not considered because of the difficulty in modeling the distribution of the inversion carriers near the junctions. Therefore, the degree of reduction in the lateral electric field is overestimated by the analytical model, as shown in Figure 9.

### 5. Conclusions

We developed a surface potential model for MOSFETs operating near the threshold voltage. The potential distribution along the channel in the inversion regime was obtained by solving the quasi-two-dimensional Poisson's equation considering the flooding of inversion carriers in the channel and junction regions. The modeling results were consistent with that obtained from numerical device simulations which indicated a transconductance overshoot near the threshold voltage. The surface potential model describes the shrinkage of the source and drain edges due to more inversion carriers flooding into the depletion regions. This widens the channel length. Therefore, the lateral electric field in the channel decreased with the gate bias, leading to an overshoot in the transconductance.

**Author Contributions:** Conceptualization, H.-C.C.; methodology, R.-D.C.; formal analysis, B.-W.L. and R.-D.C.; investigation, H.-C.C., B.-W.L. and R.-D.C.; data curation, B.-W.L., S.-Y.C. and Y.-H.H.; writing—original draft preparation, R.-D.C.; writing—review and editing, H.-C.C.; funding acquisition, H.-C.C. and R.-D.C. All authors have read and agreed to the published version of the manuscript.

**Funding:** This research was partially funded by the National Science and Technology Council of the Republic of China, Taiwan, grant numbers NSTC 111-2221-E-182-059 and NSTC 112-2221-E-182-066.

**Data Availability Statement:** The data used to support the findings of this study are available from the corresponding author upon reasonable request.

**Acknowledgments:** The authors would like to thank the National Center for High-performance Computing of the Republic of China, Taiwan, for providing the numerical simulation platform.

**Conflicts of Interest:** The authors declare no conflict of interest.

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
