# Peer review of "A Surface Potential Model for Metal-Oxide-Semiconductor Transistors Operating near the Threshold Voltage"

_electronics, doi:10.3390/electronics12204242_

Round 1

Reviewer 1 Report

This work presents a model for the surface potential of a classical MOSFET, with an emphasis on the impact of the Source and Drain regions.

1) Even though the presented model seems original to me, the following paper appears to be very similar to this work and should be cited in the state of the art: 

T. Dutta, Q. Rafhay, G. Pananakakis and G. Ghibaudo, "Modeling of the impact of source/drain regions on short channel effects in MOSFETs," 2013 14th International Conference on Ultimate Integration on Silicon (ULIS), Coventry, UK, 2013, pp. 69-72, doi: 10.1109/ULIS.2013.6523493.

The advantages of the new model over the approach of Dutta et al. should be commented.

2) The overshoot of the transconductance seems to be a short channel effect. For this reason, it would be interesting to show a corresponding Id-Vg characteristic next to Fig 3 or 4.

3) The hypothesis behind Eq. (15) should be presented. For example, do you consider accumulation, depletion, the presence of minority carriers ? Do you consider Boltzmann approximation to be valid for such a high doping ?

4) Eq. (23)  might need more explanations. It is not clear if 2D effects in the S/D regions are taken into account. 

5) A constant mobility of 1200 cm2/V/s was assumed for the modeling. Would the model still be valid if the mobility was not constant ? Is the effect studied in this paper not usually already taken into account by a gate voltage dependence on the mobility ?

There is a typo line 105: "... the electrostatics in the inversion regime of is analyzed..."

Reviewer 2 Report

General Comments: 

This work is very interesting and relevant for MDPI Electronics readership. There are a few improvements that can be made in terms of presentation and analysis. I recommend accepting this work for publication after revisions. 

Specific Comments: 

1. Is the channel widening effect (due to flooding of inversion carriers) relevant for advanced transistors like FinFET or gate-all-around nanosheets? 

2. Has the channel widening effect been discussed/analyzed before in literature? If yes, please provide references. 

3. Could the authors please add a schematic/illustration explaining how the channel widening effect occurs? 

4. Figure 8: why does the analytical model overestimate the degree of lateral electric field reduction compared to MEDICI numerical simulation?
